# METRIC-SOLVER: SLIDING ANCHORED METRIC DEPTH ESTIMATION FROM A SINGLE IMAGE

## ABSTRACT

Accurate and generalizable metric depth estimation is crucial for various computer vision applications but remains challenging due to the diverse depth scales encountered in indoor and outdoor environments. In this paper, we introduce Metric-Solver, a novel sliding anchor-based metric depth estimation method that dynamically adapts to varying scene scales. Our approach leverages an anchor-based representation, where a reference depth serves as an anchor to separate and normalize the scene depth into two components: scaled near-field depth and tapered far-field depth. The anchor acts as a normalization factor, enabling the near-field depth to be normalized within a consistent range while mapping far-field depth smoothly toward zero. Through this approach, any depth from zero to infinity in the scene can be represented within a unified representation, effectively eliminating the need to manually account for scene scale variations. More importantly, for the same scene, the anchor can slide along the depth axis, dynamically adjusting to different depth scales. A smaller anchor provides higher resolution in the near-field, improving depth precision for closer objects while a larger anchor improves depth estimation in far regions. This adaptability enables the model to handle depth predictions at varying distances and ensure strong generalization across datasets. Our design enables a unified and adaptive depth representation across diverse environments. Extensive experiments demonstrate that Metric-Solver outperforms existing methods in both accuracy and cross-dataset generalization.

## 1 INTRODUCTION

Monocular depth estimation from a single image Eigen et al. (2014); Fu et al. (2018); Bhat et al. (2021); Yuan et al. (2022); Guizilini et al. (2023); Bhat et al. (2023); Ning et al. (2023); Shao et al. (2023); Yang et al. (2024b) , plays a crucial role in a wide range of computer vision applications, including robotics Wang et al. (2025), augmented reality Kalia et al. (2019), 3D graphics Kerbl et al. (2023); Mildenhall et al. (2020) and autonomous driving Burnett et al. (2019). Depth estimation methods can be broadly divided into two types: relative depth estimation Ranftl et al. (2022); Yang et al. (2024b) and metric depth estimation Yin et al. (2023); Hu et al. (2024a); Guizilini et al. (2023); Bhat et al. (2023). Relative depth estimation methods predict the depth of objects in a scene relative to one another, providing spatial relationships between objects. In contrast, metric depth estimation aims to predict the true, real-world scale of the scene, providing accurate measurements of the distance between objects and the camera. However, metric depth estimation presents significant challenges, particularly in terms of scale variation across different datasets Bhat et al. (2023; 2021); Yuan et al. (2022), which cause metric ambiguity due to mixed-data training. Recently, many methods have been proposed to address the generalization problem in metric depth estimation by using pre-input camera intrinsics to simplify the issue, including Metric3D Hu et al. (2024a) and Depth Any Camera Guo et al. (2025), which have achieved significant performance improvements. However, for the aforementioned methods, in-the-wild images with unknown camera settings remain an challenging problem.

Another key difficulties in metric depth estimation is handling the varying depth scales across different in-the-wild scenes, such as indoor and outdoor environments. In indoor scenes, the maximum depth is typically within several meters, while in outdoor scenes, it can extend to several hundred meters. This disparity makes it challenging to use a unified normalization approach across diverse scenes, leading to issues in network training and generalization Bhat et al. (2023). Moreover, even

within the same scene or image, different regions may require different depth sensitivities, much like how humans naturally shift their focus between nearby objects and distant backgrounds depending on the task. A single fixed/learned normalization anchor cannot easily accommodate this dynamic focus, motivating the need for adaptive anchor strategies in a unified manner.

To address these challenges, we propose Metric-Solver, a novel sliding anchor-based metric depth representation that dynamically adjusts to varying depth scales. To represent depth values ranging from zero to infinity in a unified manner, we introduce a reference depth as an anchor, which partitions the scene depth into two distinct components: scaled near-field depth and tapered far-field depth, with the anchor depth acting as the normalization factor. The near-field branch captures detailed local geometry, while the far-field branch preserves distant depth information instead of discarding it entirely, helping the model better distinguish far regions such as backgrounds or sky (see Fig.1). Specifically, we adopt a *one-shared-encoder, two-lightweight-decoder* architecture, where a powerful shared encoder extracts latent features, which are then processed by two separate lightweight decoder branches to predict scaled near depthand tapered far depthrespectively, ensuring both high inference accuracy and computational efficiency. For depth values within the anchor range, we apply linear normalization to obtain the scaled near-field depth, ensuring depth values remain within a consistent range. For depth values beyond the anchor, we use an exponential normalization function to smoothly compress far-depth values, where the anchor depth is mapped to 1 and infinity gradually decays to 0. This transformation preserves depth continuity while allowing the model to handle far-field depth variations effectively. Besides, by dynamically sliding the anchor along the depth axis, our method allows the model to adaptively adjust depth scale across both inter- and intra-scene variations. For instance, a smaller anchor enhances near-field depth fidelity, improving the accuracy of predictions for closer objects, while a larger anchor better captures far-field depth relationships. This adaptability not only enhances the model's precision in different depth regions but also improves its versatility across various scene scales.

We validate our method on various benchmark datasets, including both indoor Silberman et al. (2012); Koch et al. (2018); Roberts et al. (2021); Song et al. (2015) and outdoor scenes Geiger et al. (2013); Cabon et al. (2020); Vasiljevic et al. (2019); Ros et al. (2016), to assess its robustness and generalization capability. Extensive experiments demonstrate the model's strong performance across diverse datasets and its ability to handle zero-shot settings. Fig.4 in Appendix A shows a gallery of our prediction results across various scenes. Our results emphasize the adaptability of the sliding anchor-based depth estimation approach, which consistently delivers accurate metric depth predictions across a range of scene scales.

To summarize, we present the following key contributions:

- We propose a novel framework, i.e. the sliding anchored method, which effectively models metric ambiguity in in-the-wild images with unknown camera settings.
- We design a one-shared-encoder, two-lightweight-decoder architecture that effectively bridges the varying depth scales across indoor and outdoor scenes.
- Our method demonstrates strong generalization ability across various benchmark datasets, achieving state-of-the-art (SOTA) performance on all benchmarks.

## 2 RELATED WORKS

### 2.1 RELATIVE DEPTH ESTIMATION

Relative depth estimation focuses on predicting the depth relationships between objects in a scene rather than their absolute distances. This approach is widely used to handle the high dynamic range of depth distributions in indoor and outdoor environments, where depth values can vary significantly due to differences in scene structure, lighting, and camera intrinsic parameters Yuan et al. (2022). To address these challenges, many methods Lee & Kim (2019); Ranftl et al. (2022); Birkl et al. (2023); Ranftl et al. (2021); Mertan et al. (2022) normalize depth values and use them as learning targets for neural networks. In traditional depth estimation algorithms Saxena et al. (2005); Nagai et al. (2002); Michels et al. (2005), they typically rely on dense depth regression using hand-crafted features. However, due to the limited expressiveness of these features, such methods struggle in complex environments and low-texture regions, resulting in poor depth estimation performance.

Deep-learning-based depth estimation approaches can be broadly categorized into discriminative models that employ depth regression decoders Guizilini et al. (2023); Bhat et al. (2023); Ranftl et al. (2021; 2022); Ning et al. (2023); Patil et al. (2022); Yang et al. (2024b) and generation modelsKe et al. (2024); Gui et al. (2024); Fu et al. (2024), which leverage diffusion modelsRombach et al. (2022) for depth prediction. In discriminative models, powerful backbone networks Oquab et al. (2023) extract multi-scale features from the image, and the decoder decodes the depth information. Eigen et al Eigen et al. (2014) used scale-invariant loss to train relative depth estimation networks, allowing the network to focus on predicting the relative order of pixels' depths rather than being constrained by the absolute scale of the scene. Some works improve depth estimation accuracy by introducing additional priors such as segmentation maps Zhang et al. (2018). To address scene generalization and edge blurring issues, the Depth Anything series Yang et al. (2024c; 2025) use semi-supervised strategies to complete multi-scene learning on large datasets, significantly enhancing the network's ability to refine depth at scene edges by incorporating synthetic datasets Roberts et al. (2021); Cabon et al. (2020). In generative models, Marigold Ke et al. (2024) encodes RGB images and depth separately into latent space, uses the latent code of RGB as a condition, and denoises the noisy latent code of depth. The depth is then decoded via a pretrained VAE Kingma et al. (2013), achieving accurate depth estimates. Thanks to the visual priors of pre-trained diffusion models Rombach et al. (2022), sharper depth edges are obtained. Later works such as Depth Crafter Hu et al. (2024b) and Depth Any Video Yang et al. (2024a) extend this approach to video depth generation tasks. However, relative depth estimation still faces significant limitations in downstream tasks such as environmental perception and 3D reconstruction Kerbl et al. (2023); Mildenhall et al. (2020); Yao et al. (2020) due to the absence of absolute scale information.

## 2.2 METRIC DEPTH ESTIMATION

Compared to relative depth estimation, metric depth estimation Patil et al. (2022); Ning et al. (2023); Ranftl et al. (2021); Bhat et al. (2023); Guizilini et al. (2023); Yin et al. (2023); Hu et al. (2024a) has greater application value due to the presence of precise scale information. Some approaches Yin et al. (2023); Hu et al. (2024a); Guo et al. (2025) consider the uncertainty in depth caused by camera intrinsics and introduce camera intrinsic parameters to transform the perspective image into an intrinsic-independent space for depth estimation, thereby mitigating the uncertainty introduced by camera intrinsics. However, the requirements on camera intrinsics during training or inference limit the applicability of such methods on large-scale data, such as images generated by AI or images with unknown camera intrinsics.

Therefore, some methods explore monocular metric depth estimation from a single image without requiring camera intrinsics Bhat et al. (2021; 2022; 2023); Yang et al. (2024c); Yuan et al. (2022); Lee et al. (2019); Piccinelli et al. (2024); Bochkovskii et al. (2025). While these methods achieve strong in-domain performance by overfitting to specific datasets, they typically require manually setting the maximum truncation depth for each domain. Due to limited training diversity, they struggle to generalize across indoor and outdoor scenes and lack zero-shot capability, resulting in lower accuracy for unseen environments. Additionally, some works Zhu et al. (2024) decouple the relative depth map prediction from the absolute metric scale prediction, combining them to produce the final metric depth map. Some studies Bhat et al. (2022; 2021); Shao et al. (2023) argue that direct depth regression across a large depth range is challenging, and thus, manually set maximum depths in a single scene, then, the range from 0 to the maximum depth is adaptively divided into different depth bins. Each pixel's depth value is classified, and the classification results are weighted to obtain smooth depth results. However, due to the manual setting of the maximum depth, these methods often face degraded generalization across datasets or across indoor and outdoor scenes. In this paper, we propose a sliding anchor-based representation to normalize depth from zero to infinity in a unified manner, eliminating the need for manually defining the scene's maximum depth. This enables our method to scale to larger datasets and achieve improved generalization.

## 3 METHOD

In this section, we describe our sliding anchor-based approach for monocular metric depth estimation. We first introduce the key idea behind the anchor-based depth representation (Sec. 3.1) . Then, we present the design of sliding anchor to enable accurate and scalable depth estimation across di-

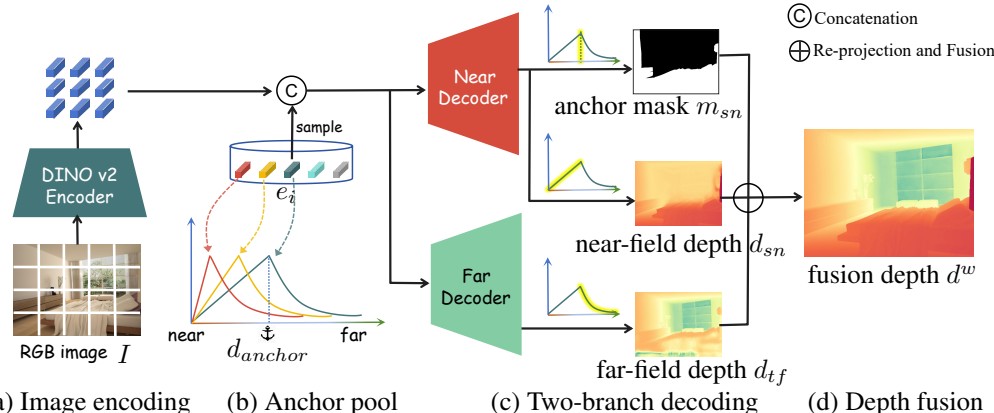

(a) Image encoding    (b) Anchor pool    (c) Two-branch decoding    (d) Depth fusion

Figure 1: **Method Overview**. Starting with an input image, (a) we use DINOv2 (Oquab et al. (2023)) to extract latent features. (b) These features are combined with a randonly sampled anchor depth from an anchor pool during training and passed into a two-branch decoder. The anchor separates near and far regions at the pixel level via an anchor mask $m_{sn}$. (c) The decoder predicts the near depth $d_{sn}$, anchor mask $m_{sn}$, and far depth $d_{tf}$. (d) The final depth is obtained by fusing the two depth maps using the mask.

verse scenes (Sec. 3.2). Finally, we outline the training loss functions used to optimize the model (Sec. 3.3). An overview of our method is shown in Fig. 1.

### 3.1 ANCHOR-BASED DEPTH REPRESENTATION

Given an input image $I$, the first step of our method is to utilize a large image encoder, to extract latent features from the input image. These features serve as a high-level representation of the image, capturing important information about the scene. Then, the encoded latent features, along with sampled anchor embedding, are processed by two decoding branches: (1) *scaled near depth* and (2) *tapered far depth*, which can formulate a complete depth prediction for the input image. Furthermore, the anchor depth dynamically adjusts along the depth axis, allowing the model to adapt to different scene scales, ensuring more flexible and accurate depth estimation and better generalization across diverse environments. Next, we will introduce each components one by one.

**Scaled Near Depth**    The first branch generates the normalized near-field depth $d_{sn}(x, y) \in [0, 1]$, where $x, y$ represent the pixel coordinates in the image. For ground truth (GT) depth $\overline{d}(x, y) \in [0, \infty]$ at pixel $(x, y)$ lies within the anchor depth $d_{anchor}$, we define the normalized near-field depth $\overline{d}_{sn} \in [0, 1]$ as:

$$\overline{d}_{sn}(x, y) = \frac{\overline{d}(x, y)}{d_{anchor}}, 0 \le \overline{d}(x, y) \le d_{anchor} \tag{1}$$

During training, we directly supervise the predicted near-field depth $d_{sn}(x, y)$ using the GT depth $\overline{d}_{sn}(x, y)$ in the corresponding near-field regions. However, beyond the reference anchor depth, ground truth supervision is absent, making predictions in these areas unreliable and unstable. This lack of supervision leads to ambiguities in distinguishing valid regions of the scaled near-depth representation during inference, reducing the model's robustness in generalization. Thus, we introduce an additional mask prediction head in this branch, as shown in Fig. 1 (c). Specifically, at the final layer of this branch, before the final depth prediction, we add a linear projection layer that maps the feature into a probability map between 0 and 1 as $m_{sn}(x, y)$, and computes the binary cross-entropy loss against the GT valid mask $\overline{m}_{sn}(x, y)$. During inference, we apply a threshold of 0.5 to obtain the binary mask. This binary mask indicates the valid areas of the scaled depth prediction: the valid areas are assigned a value of 1 (true), and the invalid areas are assigned a value of 0 (false). By incorporating this mask prediction, the model can effectively differentiate between valid and invalid depth regions, leading to more stable and reliable depth estimation. Formally, we define the GT valid mask $\overline{m}_{sn}(x, y)$ for the scaled near-depth branch as:

$$\overline{m}_{sn}(x,y) = \mathbb{I}(\overline{d}(x,y) \leq d_{anchor}) \tag{2}$$

where $\mathbb{I}(\cdot)$ is the indicator function.

By incorporating this mask prediction mechanism, we ensure that both branches focus on the most reliable parts of the scene, resulting in more accurate and robust depth predictions.

**Tapered Far Depth**    The second branch generates the tapered far-field depth $d_{\text{tf}}(x,y) \in [0,1]$, which captures depth information beyond the anchor depth $d_{\text{anchor}}$. The corresponding depth range is $[d_{\text{anchor}}, \infty]$. In previous methods (e.g., ZoeDepthBhat et al. (2023)), depth values exceeding the maximum normalization threshold are often discarded during training, resulting in a loss of valuable geometric information at test time. For example, this can cause the model to misinterpret distant regions, such as failing to distinguish between actual far-field structures and background sky.

To normalize these far-field GT depths $\overline{d}(x,y)$ into a consistent range, we apply an exponential normalization function with a negative exponent, ensuring that the anchor depth is mapped to 1, while far depths gradually decay to 0 in a smooth and continuous manner. The normalization function is defined as:

$$\overline{d_{\text{tf}}}(x,y) = e^{-k(\overline{d}(x,y) - d_{\text{anchor}})}, \overline{d}(x,y) \geq d_{anchor} \tag{3}$$

where $k$ is a hyperparameter that controls the rate of depth attenuation beyond the reference anchor depth. In our implementation, we set $k = 0.025$ to achieve a smooth and stable depth transition.

**Depth Re-projection and Fusion**    After the two branches generate normalized depth representations, they are reprojected into real-world depth values and fused to obtain a complete depth prediction in real-world scale, denoted as $d^w(x,y) \in [0,\infty]$. First, we compute the inverse transformations of $d_{sn}$ and $d_{tf}$ to get $d_{sn}^w$ and $d_{tf}^w$ in real-world scale using the reference anchor $d_{\text{anchor}}$ as the scaling factor.

$$d_{sn}^w(x,y) = d_{sn}(x,y) \cdot d_{anchor} \tag{4}$$

$$d_{tf}^w(x,y) = \frac{-\ln d_{tf}(x,y)}{k} + d_{anchor} \tag{5}$$

Then, these two depth components are fused using the mask $m_{sn}(x,y)$:

$$d^w(x,y) = m_{sn}(x,y) \cdot d_{sn}^w(x,y) + (1 - m_{sn}(x,y)) \cdot d_{tf}^w \tag{6}$$

### 3.2 Sliding Anchor

The core idea of our representation is to use a reference depth as an anchor to normalize depth from zero to infinity in a unified manner. To achieve robust generalization across varying scene scales, we construct an anchor pool that spans from near to far, consisting of a set of learnable embeddings:

$$E_{\text{anchor}} = \{e_1, e_2, ..., e_N\} \tag{7}$$

where $e_i$ represents the embedding corresponding to a specific anchor depth $d_{\text{anchor},i}$, and $N$ is the total number of anchor depths in the pool.

During training, we randomly sample an anchor depth $d_{\text{anchor}}$ for each input image and use its corresponding embedding $e_{\text{anchor}}$ to modulate the depth normalization process. We expand $e_{\text{anchor}}$ to match the spatial size of the DINO feature map and concatenate it channel-wise with the feature, which then serves as the inputs of the near and far decoders (See Fig. 1 (b)). This approach allows the model to learn how to adjust depth predictions based on varying scene scales, significantly improving generalization across diverse environments. At inference time, the choice of anchor embedding allows us to control the network's focus on different depth ranges. Specifically: (1) Near-field embedding: If a smaller anchor depth is selected, the network prioritizes closer objects, yielding higher resolution depth estimates for the near-field. This is because a smaller $d_{\text{anchor}}$ increases the effective resolution within the [0,1] normalized range. (2) Far-field embedding: If a larger anchor depth is used, the model extends its focus to distant objects, better capturing depth variations in far regions.

By leveraging anchor embeddings, our approach enables the model to adaptively adjust depth estimation across inter- and intra- scenes with different depth scales, improving accuracy and flexibility in real-world applications. For further discussion on the use of sliding anchors from the perspective of network capacity, please refer to Appendix D.

### 3.3 TRAINING

During training, we optimize the model using three distinct loss components derived from both branches, including a depth loss for each branch and a mask loss to distinguish valid regions.

**Scaled Near Depth Loss**  The scaled near depth loss, denoted as $\mathcal{L}_{sn}$, measures the difference between the predicted scaled near depth $d_{sn}(x, y)$ and the ground truth $\overline{d}_{sn}(x, y)$. We use the $L_2$ loss to compute this difference:

$$\mathcal{L}_{sn} = \sum_{x,y} \left( d_{sn}(x, y) - \overline{d}_{sn}(x, y) \right)^2 \cdot \overline{m}_{sn}(x, y) \tag{8}$$

**Tapered Far Depth Loss**  Similarly, the tapered far depth loss, denoted as $\mathcal{L}_{tf}$, measures the difference between the predicted depth $d_{tf}(x, y)$ and the ground truth depth $\overline{d}_{tf}(x, y)$:

$$\mathcal{L}_{tf} = \sum_{x,y} \left( d_{tf}(x, y) - \overline{d}_{tf}(x, y) \right)^2 \cdot (1 - \overline{m}_{sn}(x, y)) \tag{9}$$

**Scaled Near Depth Mask Loss**  The mask loss for the scaled depth branch, denoted as $\mathcal{L}_{mask}$, is computed using binary cross-entropy (BCE) loss between the predicted mask $m_{sn}(x, y)$ and the ground truth $\overline{m}_{sn}(x, y)$. The BCE loss can be expressed as:

$$\mathcal{L}_{mask} = \sum_{x,y} \text{BCE}(m_{sn}(x, y), \overline{m}_{sn}(x, y)) \tag{10}$$

**Total Loss**  The total loss for the model is the sum of the three losses from both branches:

$$\mathcal{L}_{\text{total}} = \lambda_{sn} * \mathcal{L}_{sn} + \lambda_{tf} * \mathcal{L}_{tf} + \lambda_m * \mathcal{L}_{mask} \tag{11}$$

where $\lambda_{sn}$, $\lambda_{tf}$, and $\lambda_m$ are set to 1.0, 1.0, and 0.05, respectively, to balance the contributions of the three corresponding loss components, respectively.

This comprehensive loss function allows the model to jointly optimize for accurate depth predictions and reliable mask predictions, ensuring that the model focuses on valid regions of the depth map while making robust predictions across varying scene scales.

## 4 EXPERIMENTS

### 4.1 IMPLEMENTATION DETAILS

Our model architecture consists of an image encoder and two decoding branches. The first branch predicts normalized real depth, while the second branch predicts normalized reversed depth. Both branches are adapted from the Dense Prediction Transformer (DPT) Ranftl et al. (2021) head to align with our sliding anchor-based approach. For the image encoder, we use DINOv2 Oquab et al. (2023), same as DepthAnythingV2 large Yang et al. (2025). The DPT heads are randomly initialized to learn depth predictions tailored to our method. The training is conducted using the AdamW Loshchilov & Hutter (2017) optimizer with a learning rate of $5 \times 10^{-6}$, running on 8 NVIDIA H100 GPUs for a total of 205K steps. And the model is pretrained on a combination of diverse datasets covering a wide range of scene scales. Please refer to Appendix C for more details.

### 4.2 COMPARISONS

**Zero-shot Generalization**  To test the generalization performance of different models in zero-shot indoor and outdoor scenarios, we conducted performance comparisons in 4 datasets: 2 indoor datasets (iBims Koch et al. (2018) and DIODE Vasiljevic et al. (2019) Indoor) and 2 outdoor datasets (DIODO Vasiljevic et al. (2019) Outdoor and SYNTHIA Ros et al. (2016)). As shown in Table 1, our model achieves the best generalization performance across all datasets. This further demonstrates the effectiveness of our design.

| Method | iBims | | | DIODE Indoor | | | DIODE Outdoor | | | SYNTHIA | | |
|--------|-------|------|------|------|------|------|------|------|------|------|------|------|
| | $\delta_1\uparrow$ | REL↓ | RMSE↓ | $\delta_1\uparrow$ | REL↓ | RMSE↓ | $\delta_1\uparrow$ | REL↓ | RMSE↓ | $\delta_1\uparrow$ | REL↓ | RMSE↓ |
| AdaBins | 0.555 | 0.212 | 0.901 | 0.174 | 0.443 | 1.963 | 0.161 | 0.863 | 10.35 | 0.832 | 0.350 | 6.271 |
| BTS | 0.538 | 0.231 | 0.919 | 0.210 | 0.418 | 1.905 | 0.171 | 0.837 | 10.48 | 0.863 | 0.785 | 4.920 |
| LocalBins | 0.558 | 0.211 | 0.880 | 0.229 | 0.412 | 1.853 | 0.170 | 0.821 | 10.27 | 0.901 | 0.720 | 5.707 |
| NeWCRFs | 0.548 | 0.206 | 0.861 | 0.187 | 0.404 | 1.867 | 0.176 | 0.854 | 9.228 | 0.923 | 0.468 | 5.934 |
| ZoeDepth | 0.612 | 0.185 | 0.732 | 0.247 | 0.371 | 1.842 | 0.269 | 0.852 | 6.898 | 0.912 | 0.413 | 4.762 |
| DAV2 | 0.512 | 0.243 | 0.848 | 0.311 | 1.511 | 6.774 | 0.192 | 1.435 | 10.14 | 0.936 | 0.325 | 4.934 |
| ZoeDepth-NK | 0.588 | 0.192 | 0.830 | 0.386 | 0.331 | 1.598 | 0.208 | 0.757 | 7.569 | 0.902 | 0.824 | 4.274 |
| Unidepth | 0.541 | 0.193 | 0.752 | 0.278 | 0.479 | 1.741 | 0.235 | 0.781 | 7.421 | 0.893 | 0.475 | 5.329 |
| DepthPro | 0.762 | 0.181 | 0.527 | 0.401 | 0.363 | 1.462 | **0.391** | **0.613** | **4.712** | 0.931 | 0.341 | 4.515 |
| **ours** | **0.910** | **0.111** | **0.409** | **0.446** | **0.279** | **1.180** | 0.383 | 0.656 | 4.836 | **0.966** | **0.153** | **3.842** |

Table 1: **Zero-shot generalization on indoor and outdoor datasets.** Note that methods in the top block employ separate models for indoor and outdoor scenes, while methods in the bottom block use a single unified model for inference across all datasets. Our unified model achieves the best or near-best performance across all datasets, demonstrating the generalization capability.

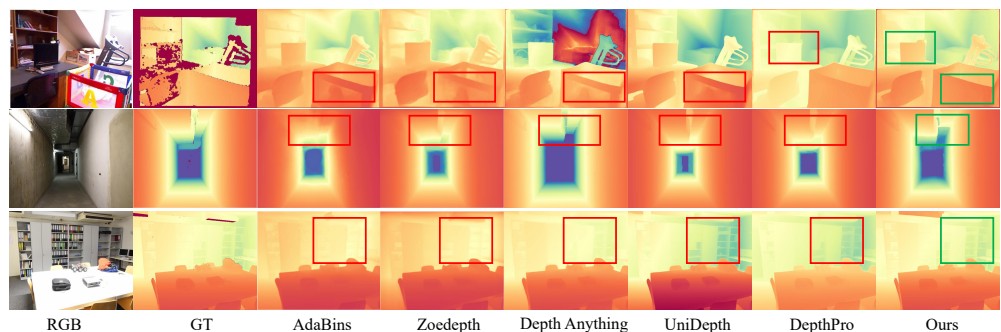

RGB GT AdaBins Zoedepth Depth Anything UniDepth DepthPro Ours

Figure 2: **Qualitative comparisons on the indoor dataset**. When dealing with large-scale and long-distance indoor scenes, our framework achieves better absolute depth recovery.

**Comparison in NYU and KITTI** Since most metric depth estimation methods are fine-tuned separately on NYU-V2 Silberman et al. (2012) and KITTI Geiger et al. (2013), these datasets serve as the primary benchmarks for evaluating indoor and outdoor performance. We finetune the pretrained model on these two datasets for 70 epochs and compare our method against state-of-the-art (SOTA) monocular depth estimation models on NYU-V2 (indoor) and KITTI (outdoor). Because other methods impose a maximum depth threshold, we limit our evaluation depth range to 10m for NYU-V2 and 80m for KITTI to ensure a fair comparison.

As shown in Table 2, our single model outperforms the baselines on both indoor and outdoor datasets (i.e., the second block in each sub-table), even when some baselines are fine-tuned on each dataset individually (i.e., the first block in each sub-table). This improvement is attributed to our joint training across multiple datasets, which enhances our model's generalization ability across diverse depth distributions. The qualitative visualization experiments can be seen in Fig. 2 and 8 (in Appendix F) for indoor scenes and outdoor street scenes, respectively. It can be observed that our approach not only captures details more accurately and achieves the lowest overall error but also demonstrates the farthest prediction range on the KITTI dataset. Additionally, it effectively handles complex scenarios such as the sky. See Appendix F for more visual results on additional datasets.

## 4.3 ANALYSIS

**Ablation** To ablate the significance of our design choices, we conduct experiments using training data from vkitti Cabon et al. (2020) and HyperSim Roberts et al. (2021), respectively. We test three different settings to assess the impact of each component: (1) One head: The architecture employs a single decoder head, following the same design as DepthAnything, and the output depth is limited to a specific range (i.e., 80 meters to accommodate both indoor and outdoor environments). (2) Without mask head: In this setup, we use the reference anchor depth to truncate the predictions and fuse the predictions naively, without the additional mask prediction head. (3) Two decoder head with

| Method | Higher is better ↑ | | | Lower is better ↓ | | |
|---|---|---|---|---|---|---|
| | $\delta_1$ | $\delta_2$ | $\delta_3$ | AbsRel | RMSE | log10 |
| AdaBins-N | 0.903 | 0.984 | 0.997 | 0.103 | 0.364 | 0.044 |
| NeWCRFs-N | 0.954 | 0.981 | 0.997 | 0.113 | 0.394 | 0.083 |
| P3Depth | 0.898 | 0.981 | 0.996 | 0.104 | 0.356 | 0.043 |
| SwinV2 | 0.949 | 0.994 | 0.999 | 0.083 | 0.287 | 0.035 |
| IEBins | 0.936 | 0.992 | 0.998 | 0.087 | 0.314 | 0.038 |
| ZoeDepth-N | 0.955 | 0.995 | 0.999 | 0.075 | 0.269 | 0.032 |
| DAV1-N | 0.984 | 0.998 | **1.000** | 0.056 | 0.205 | 0.024 |
| ZoeDepth-NK | 0.952 | 0.995 | 0.999 | 0.077 | 0.280 | 0.033 |
| Ours | **0.986** | **0.998** | 0.999 | **0.049** | **0.183** | **0.021** |

(a) Comparison on NYU-D dataset

| Method | Higher is better ↑ | | | Lower is better ↓ | | |
|---|---|---|---|---|---|---|
| | $\delta_1$ | $\delta_2$ | $\delta_3$ | AbsRel | RMSE | log10 |
| AdaBins-K | 0.944 | 0.991 | 0.998 | 0.071 | 3.039 | 0.031 |
| NeWCRFs-K | 0.974 | 0.997 | 0.999 | 0.052 | 2.129 | 0.079 |
| P3Depth | 0.953 | 0.993 | 0.998 | 0.071 | 2.842 | 0.103 |
| SwinV2 | 0.977 | 0.998 | 1.000 | 0.050 | 1.966 | 0.075 |
| IEBins | 0.976 | 0.997 | 0.999 | 0.048 | 2.044 | 0.076 |
| NDDepth | 0.978 | 0.998 | 0.999 | 0.050 | 2.025 | 0.075 |
| DAV1-K | **0.982** | 0.998 | 1.000 | **0.046** | **1.896** | **0.024** |
| ZoeDepth-NK | 0.971 | 0.996 | 0.999 | 0.054 | 2.281 | 0.082 |
| Ours | 0.976 | **0.998** | **1.000** | 0.052 | 2.281 | 0.031 |

(b) Comparison on KITTI dataset

Table 2: **Quantitative in-domain metric depth estimation**. All compared methods use the encoder size close to ViT-L. Each sub-table consists of two blocks: the first block shows results from methods fine-tuned on a specific domain (indoor or outdoor), while the second block presents results from methods fine-tuned jointly on both domains. The model names suffixed with "N" indicate models specifically optimized for NYU, while those with "K" denote models optimized for KITTI.

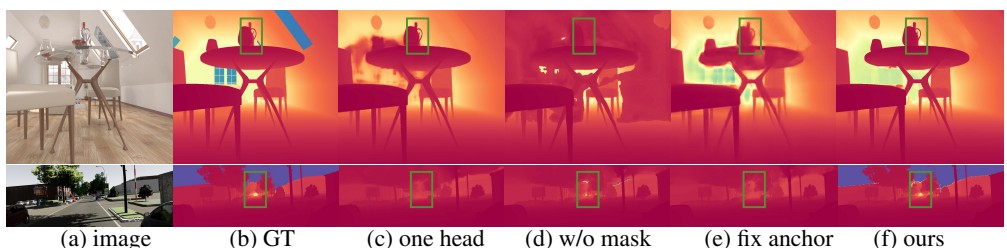

| (a) image | (b) GT | (c) one head | (d) w/o mask | (e) fix anchor | (f) ours |

Figure 3: **Qualitative comparisons of different ablation settings**. Compared with the baseline settings (c), our full setting (f) allows for effective observing further distances (e.g., sky in the second row). And the anchor mask-based fusion strategy ensures seamless stitching of near and far depths (d) and higher depth fidelity in near-range indoor scenes (e) in indoor scenes.

fixed anchor: We adopt only one anchor to validate the significance of the sliding anchor design. (4) Our full setting: This is the full configuration, including both branches with mask prediction and anchor depth injection.

Fig. 3 illustrates the qualitative performance. As shown in Fig. 3 (c) (second row), the One-head model fails to predict depths beyond the predefined maximum depth, leading to information loss. Removing the mask head cannot produce accurate fusion of the two depth branches (Fig.3 (d)), while using a fixed anchor results in the loss of fine-grained details (Fig.3 (e)). Table.3a further presents the quantitative results. Our full setting achieves the best performance across all metrics. These results demonstrate the effectiveness of the sliding anchor-based depth representation and the importance of the mask prediction mechanism.

| Model setting | $\delta_1$↑ | $\delta_2$↑ | REL↓ | RMSE↓ | log10↓ |
|---|---|---|---|---|---|
| one head | 0.471 | 0.698 | 0.363 | 3.525 | 0.176 |
| w/o mask head | 0.341 | 0.813 | 0.563 | 10.324 | 0.321 |
| fix-anchor | 0.701 | 0.917 | 0.200 | 2.744 | 0.085 |
| Ours (full) | **0.734** | **0.935** | **0.189** | **2.616** | **0.071** |

(a)

| Anchor | 20m | | 80m | | 120m | |
|---|---|---|---|---|---|---|
| | $\delta_1$↑ | REL↓ | $\delta_1$↑ | REL↓ | $\delta_1$↑ | REL↓ |
| Anchor-1(20m) | **0.963** | **0.066** | 0.871 | 0.128 | 0.771 | 0.154 |
| Anchor-2(40m) | 0.960 | 0.068 | 0.917 | **0.092** | 0.804 | 0.147 |
| Anchor-3(80m) | 0.953 | 0.071 | **0.929** | 0.096 | 0.851 | 0.155 |
| Anchor-4(120m) | 0.936 | 0.086 | 0.916 | 0.108 | **0.908** | **0.121** |

(b)

Table 3: **Analysis**. (a) Ablation studies of model design. (b) Evaluation of different anchor in vkitti.

**Anchor Embeddings** To evaluate the impact of anchor embeddings, we conduct experiments on the vkitti Cabon et al. (2020) dataset, using consistent input data while varying anchor embeddings as depth prediction conditions. Performance is evaluated across different maximum truncation depths. As shown in Table 3b and Fig. 7 in Appendix B, the model achieves the best results when the truncation depth matches the anchor depth, indicating that it effectively adapts to different anchor embeddings and produces optimal predictions when evaluated within the corresponding depth range. Notably, the model achieves the highest accuracy with the smallest anchor embedding, demonstrat-

ing that a smaller anchor improves depth precision for closer objects. Fig. 6 in Appendix B presents a visualization of the impact of different anchor embeddings, illustrating that as the anchor depth increases, the valid prediction region (i.e., the fusion mask) of the scaled depth branch expands accordingly. For more analysis on mask-based depth fusion and mask accuracy under different anchors, please refer to Appendix B.

**Memory Consumption and Efficiency** We evaluate the efficiency of our method against DepthAnything using the same DINOv2 backbone on 512×512 input images with an H100 GPU (see Table 4). Our dual-decoder design adds only one additional DPT head with 58M parameters, while sharing the same backbone features that dominate inference time. As a result, the runtime increase is minimal (approximately 4ms), thanks to the lightweight nature of the DPT head. Overall, our dual-decoder method introduces just 8ms time cost, enabling a real-time sliding anchor mechanism and supporting inference speeds up to 125 FPS ($= 1sec/8ms$),

| Methods | Paras | Time | Memory |
|---|---|---|---|
| DepthAnything(L) | 335.3M | 28.0ms | $1,820$ MB |
| Ours | 393.0M | 32.2ms | $2,151$ MB |

Table 4: Model efficiency and resource consumption.

**Anchor Selection Strategy During Inference** Our method supports flexible anchor selection at inference time, enabling dynamic adaptation to various tasks, scenes, and application needs. (1) For evaluation comparisons, we match the anchor value to the baseline truncation values to ensure fairness, using small anchor (10m) and large anchor (80m) to evaluate indoor and outdoor scenes, respectively; (2) For downstream applications, our method supports the selection of any anchor to generate accurate metric depth values ranging from 0 to infinity according to the task requirements. Benefitting from the efficiency of our light-weight decoder design (See Table 3b), our method supports three anchor selection strategies: (1) task-specific selection (e.g., small anchors for indoor precision, large ones for outdoor range), (2) multi-anchor fusion to improve depth accuracy with low latency, and (3) dynamic adjustment based on semantic cues from vision-language models. Further detailed discussions are provided in Appendix E.

An alternative approach is to predict the anchor scale directly from global image features. However, this typically produces a single global anchor and lacks the flexibility to adapt to varying depth distributions within the same scene. In contrast, our method introduces a sliding anchor mechanism that provides a sliding "anchor bar" and allows users or downstream tasks to flexibly shift attention from near to far regions based on task/focus demands. Moreover, this interactive process can run in real time at 125 FPS. This flexibility enables more adaptable inference across diverse settings.

## 5 CONCLUSION AND LIMITATIONS

In this paper, we proposed a sliding anchor-based metric depth estimation method to address the challenges of scale variation across diverse environments. Our dynamic sliding anchor allows the model to adapt to varying depth scales, ensuring precise predictions in both near and far fields. Our framework leverages a pretrained DINOv2 encoder and two modified DPT heads for depth prediction while incorporating learnable anchor embeddings to seamlessly encode depth reference information. Additionally, we introduced a mask prediction mechanism to enhance the robust fusion of depth predictions from two branches, improving model stability and generalization across datasets. Extensive experiments demonstrate that our method outperforms existing approaches on both indoor and outdoor benchmarks, achieving strong generalization without relying on scene-specific assumptions such as fixed maximum depths or prior knowledge of the scene type. Our findings suggest that the proposed sliding anchor-based representation offers a scalable and effective solution for metric depth estimation across a wide range of real-world applications.

**Limitations** Metric-Solver is currently designed for single-image input. Extending it to monocular video metric depth estimation is a meaningful direction for enabling broader and more diverse applications. Another limitation lies in handling challenging regions such as textureless surfaces and transparent areas, where depth cues are inherently ambiguous or unreliable. We leave these issues as important directions for future work.

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

## A RESULT GALLERY

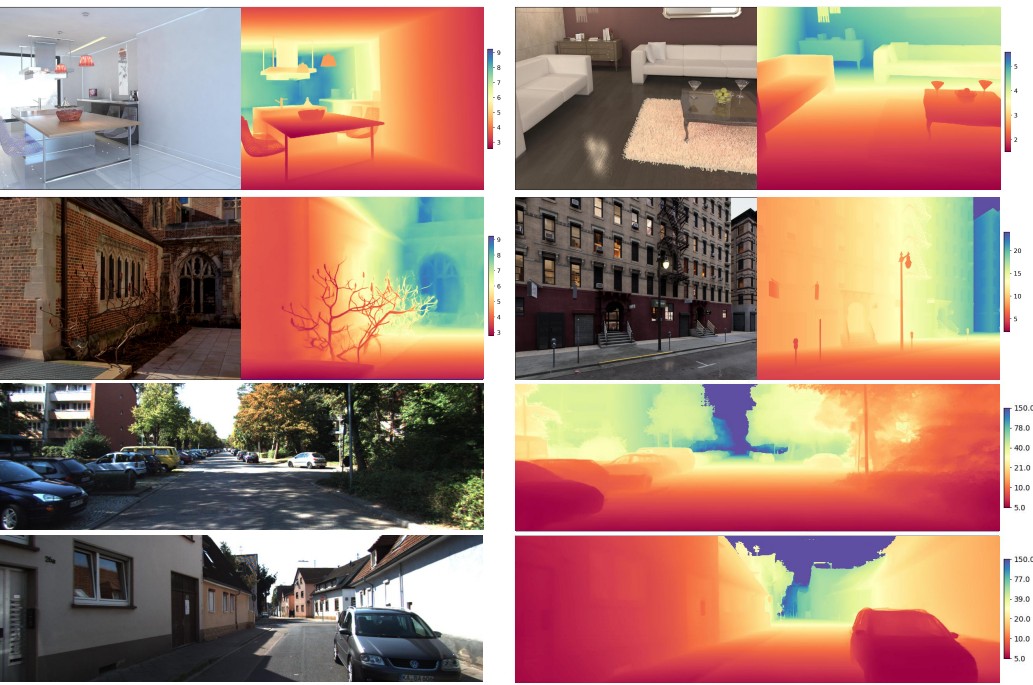

Figure 4: **A gallery of our predictions across various scenarios.** The Metric-Solver model effectively addresses different in-the-wild scenes with unknown camera settings. This model delivers precise metric depth predictions across a variety of scenarios, including but not limited to indoor and outdoor scenes, autonomous driving scenarios, and various datasets which are captured by different cameras. The side bar along each depth map indicates the predicted depth range in meters.

Figure 4 presents a gallery of depth estimation results of our method under different working conditions. Our method can well handle absolute depth estimation and detail preservation in various scenarios. In the supplementary document, we mainly provide the following contents: (1) the accuracy evaluation of mask prediction, as well as the mask prediction accuracy and depth accuracy under different anchors; (2) relevant training data; (3) the theoretical analysis on the effectiveness of sliding anchors; (4) more visualized comparisons on indoor and outdoor test datasets.

## B FUSION MASK ANALYSIS

### B.1 ACCURACY OF FUSION MASK

The table 5 presents the pixel accuracy of mask predictions using different depth anchor embeddings across two datasets. The purpose of this table is to validate the effectiveness of the mask fusion strategy. Specifically, it demonstrates how varying depth anchor embeddings impact the accuracy of mask predictions, thus evaluating the performance of the fusion strategy in different depth scenarios. The pixel accuracy is computed by comparing the predicted mask with the ground truth (GT) mask at the pixel level. The formula for pixel accuracy is as follows:

$$\text{Pixel Accuracy} = \frac{\text{Number of Correctly Predicted Pixels}}{\text{Total Number of Pixels}}$$

Here, the "correctly predicted pixels" refer to the pixels in the predicted mask that match the corresponding pixels in the GT mask.

| Evaluation Metric | Pixel Accuracy for Different Depth Anchor Embeddings | | | | | | | | |
|---|---|---|---|---|---|---|---|---|---|
| Setting | H2m | H4m | H6m | H10m | H20m | V10m | V20m | V40m | V80m | V120m |
| Accuracy | 0.9473 | 0.8935 | 0.8970 | 0.9297 | 0.9594 | 0.9704 | 0.9670 | 0.9714 | 0.9847 | 0.9868 |

Table 5: Mask Prediction Accuracy Results of Different Depth Anchor Embeddings on Two Datasets. Note: H stands for the Hypersim dataset, V stands for the VKITTI dataset; the numbers after H/V represent depth values in meters. For example, H5m means the 5-meter depth anchor embedding of the Hypersim dataset

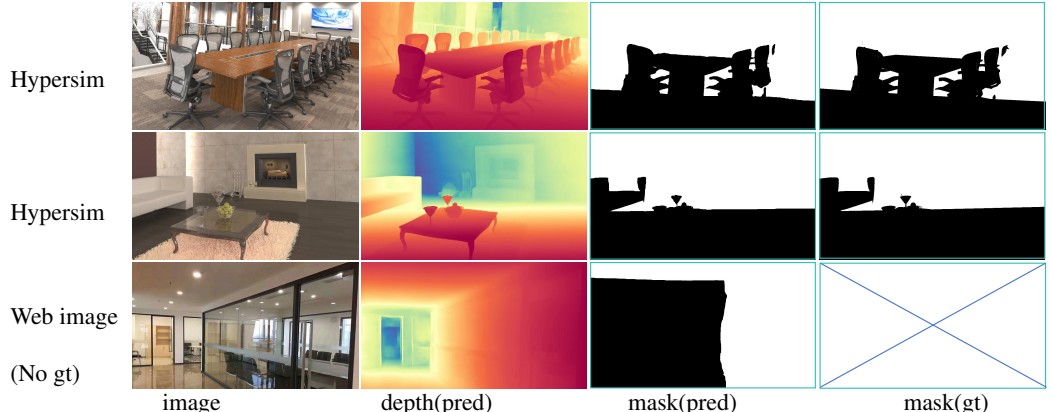

Figure 5: **Mask prediction visualization.** By learning on large scale data, our mask prediction head performs robustly across various scenes.

Meanwhile, to verify the accuracy and robustness of the fusion mask prediction, the corresponding anchor masks and actual masks were tested under various working conditions. The visualized comparison is shown in Figure 5, and our mask mechanism can accurately handle various scenarios.

### B.2 ACCURACY OF DIFFERENT ANCHOR

To verify the performance gain under different anchors, we not only completed the verification on the outdoor dataset (as shown in Table 3b) but also conducted tests on the indoor dataset, which is presented in Table 6. The basic conclusion is consistent with that of the outdoor scenario: when greater emphasis is placed on the depth accuracy of nearby regions, a closer anchor can be used. Meanwhile, as illustrated in Figure 6, we show the output of the near-field head and the corresponding fusion mask when different anchors are used in the outdoor scenario.

The advantage of flexibly setting the focus range by sliding anchors and improving absolute depth is demonstrated in Fig. 7. Our model can achieve robust predictions across various anchors. Meanwhile, when we only focus on the absolute depth within the range of approximately 20 meters, using a 20-meter anchor can significantly enhance the depth accuracy within this range.

| | 2m | | 4m | | 10m | |
|---|---|---|---|---|---|---|
| Anchor | $\delta_1 \uparrow$ | REL $\downarrow$ | $\delta_1 \uparrow$ | REL $\downarrow$ | $\delta_1 \uparrow$ | REL $\downarrow$ |
| Anchor-1(2m) | **0.771** | **0.176** | 0.760 | **0.173** | 0.708 | 0.204 |
| Anchor-2(4m) | 0.762 | 0.182 | **0.763** | 0.176 | 0.712 | 0.201 |
| Anchor-3(6m) | 0.759 | 0.193 | 0.734 | 0.182 | 0.716 | 0.198 |
| Anchor-4(10m) | 0.761 | 0.189 | 0.731 | 0.201 | **0.735** | **0.184** |

Table 6: **Evaluation of different anchor in HyperSim dataset**.

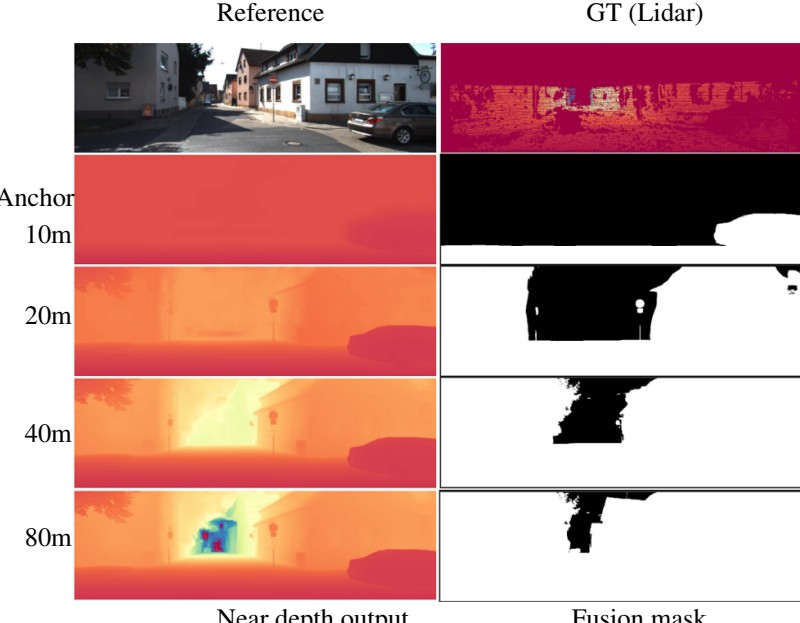

Figure 6: **Qualitative comparisons of different reference anchor depth.**It can be observed that anchors at different distances allow the near head to precisely focus on depths within different ranges and provide accurate anchor depth masks.

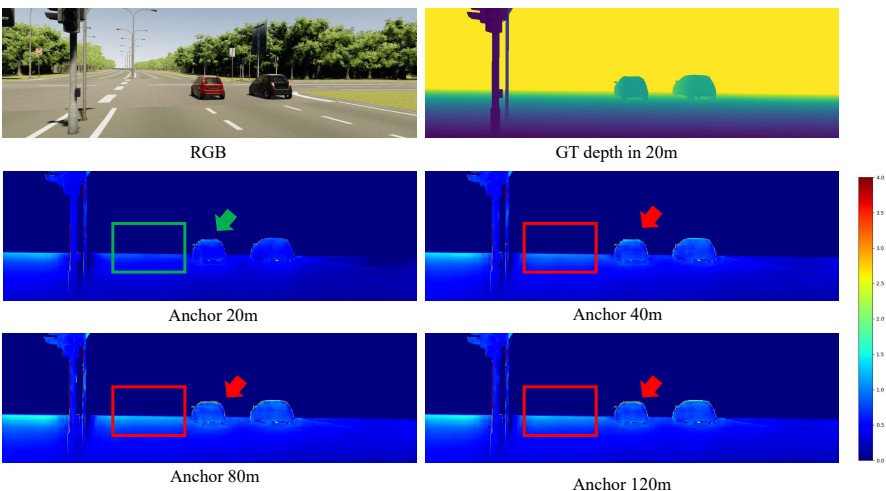

Figure 7: **Absolute Error Map of Depth Estimation for Near-Field Scenes Under Different Anchors**.

## C DATASETS

We train our model on a diverse set of both real and synthetic datasets that span a variety of ranges and scenes, as listed in Table 7. By leveraging this extensive training data, our model is able to effectively capture the complexities of different environments. As a result, we achieve depth map estimations that not only have a high dynamic range but also exhibit sharp, well-defined edges. This allows for more accurate and reliable depth perception across various contexts, enhancing the overall quality of 3D reconstruction and scene understanding.

| Datasets | Scenes | Depth From | Pair Size |
|----------|--------|------------|-----------|
| ETH3D | outdoor | real | 454 |
| DSEC | outdoor | real | 63931 |
| vkitti | outdoor | synthetic | 20000 |
| MVS-Synth | outdoor | synthetic | 12000 |
| hypersim | indoor | synthetic | 142350 |
| DIML | out/indoor | real | 70030 |
| cityscapes | outdoor | real | 174998 |

Table 7: **Datasets for training.** We train our model on real and synthetic datasets across different ranges and scenes, achieving high dynamic range and sharp-edged depth map estimation.

## D  NETWORK-CAPACITY VIEW OF SLIDING-ANCHOR NORMALIZATION

Many depth-estimation networks constrain the final output to a fixed numeric range (e.g., $[0, 1]$). Under finite representational capacity—effectively $K$ distinguishable output levels due to numerical precision, gradients, and calibration of the network—the *metric* resolution depends on how depth is mapped into this range. A global linear mapping over a large span $[0, Z_{\max}]$ allocates a uniform step size of roughly $Z_{\max}/K$ meters per level. A *sliding-anchor* mapping can adjust capacity by concentrating resolution within an adaptive anchor $a$ range. Intuitively, this mimics human focus shift, placing fine detail where the task needs it.

**Formulation (Near-Field Linear Normalization)**  Let $z > 0$ be metric depth and $a > 0$ an anchor. Focusing on the near field, we normalize depth linearly:

$$y(z; a) = \frac{z}{a}, \qquad z \le a, \ y \in [0, 1].$$

The inverse mapping used at inference/evaluation time is

$$z(y; a) = a\, y.$$

**Effective Metric Resolution**  If the network effectively provides $K$ distinguishable output levels (granularity $\Delta y \approx 1/K$), then the metric resolution at depth $z$ is

$$\Delta z(z) \ \approx \ \frac{\Delta y}{\frac{dy}{dz}}.$$

For the near-field linear mapping, $\frac{dy}{dz} = \frac{1}{a}$, hence

$$\Delta z \ \approx \ \frac{a}{K}.$$

**Comparison to Fixed Global Linear Normalization**  Methods that rely on either a fixed maximum normalization depth for different datasets, or a learnable normalization depth trained across datasets, ultimately produce a single global normalization depth. This means that during inference, a test image is normalized using a fixed global depth range, regardless of image-specific depth variations or task-specific requirements.

With a fixed global range $[0, Z_{\max}]$, $y = z/Z_{\max}$ so $\frac{dy}{dz} = \frac{1}{Z_{\max}}$ and

$$\Delta z \ \approx \ \frac{Z_{\max}}{K} \quad .$$

In contrast, using a near-field anchor $a$ yields $\Delta z \approx a/K$, enabling finer resolution when $a < Z_{\max}$. For example, if finer resolution is required in the near field during inference, a smaller anchor can be

adopted. Conversely, a larger anchor can be used when a more balanced prediction across the depth range is desired, as demonstrated in Table 1 in the main paper. This demonstrates the flexibility and adaptability of our method in accommodating different resolution requirements across depth ranges.

Summary Sliding-anchor normalization provides a principled, zoom-like mechanism to reallocate network capacity along the depth axis. It enables high near-field precision and far-field stability within a unified representation, and—by sliding the anchor—adapts to diverse scenes and camera settings without changing the model's output range.

## E  ANCHOR SELECTION STRATEGY

In the evaluation experiments, we adopt anchor depths that match the maximum normalization depths used by baseline methods to ensure fair comparisons. However, for downstream applications, a dedicated inference strategy is required to fully leverage the flexibility of our approach. Below, we outline three representative anchor selection strategies with detailed discussions:

(1) Task-specific selection: the anchor can be chosen based on the specific resolution requirements of the task. For example, a smaller anchor enables finer resolution in the near field, which is beneficial for indoor navigation, while a larger anchor provides more balanced coverage, suitable for outdoor scene understanding. Benefiting from our unified model and shared representations, such anchor adaptation can be seamlessly applied without retraining or architecture changes.

(2) Multi-anchor fusion: predictions from multiple anchors can be fused to improve robustness across the entire depth range. As discussed in the efficiency analysis (see Table 4), our lightweight decoder design allows each inference to run in only $8ms$. This enables efficient multi-anchor inference: for example, using four anchors sequentially takes just $32ms$, or only $8ms$ if processed in parallel. Such fusion not only preserves efficiency but also significantly enhances the quality of reconstruction results by leveraging complementary depth cues from different anchors.

(3) Agent-based selection: when integrated with a vision-language model (VLM), the system can dynamically adjust the anchor based on semantic cues, focusing on either near or far regions depending on the context of the same scene. For example, in a robotic manipulation task, the agent may initially adopt a larger anchor to obtain a global overview of the scene, and later switch to a smaller anchor to focus on the near field when executing fine-grained actions.

## F  MORE VISUALIZATION RESULTS

We further present visual comparisons on both indoor and outdoor datasets in Fig.8, Fig.9, and Fig. 10. Our method not only maintains high depth accuracy but also preserves fine structural details, resulting in high-quality scene predictions. Moreover, it generalizes well across diverse environments using a single model, demonstrating strong adaptability and robustness. This combination of depth fidelity, detail preservation, and cross-scene generalization distinguishes our approach in terms of both efficiency and effectiveness.

## G  EVALUATION METRICS

We evaluate depth prediction in metric depth space $\mathbf{d}$ using several standard metrics. Specifically, we compute:

Absolute Relative Error (REL):

$$\text{REL} = \frac{1}{M} \sum_{i=1}^{M} \frac{|\mathbf{d}_i - \hat{\mathbf{d}}_i|}{\mathbf{d}_i}$$

Root Mean Squared Error (RMSE):

$$\text{RMSE} = \left[ \frac{1}{M} \sum_{i=1}^{M} |\mathbf{d}_i - \hat{\mathbf{d}}_i|^2 \right]^{\frac{1}{2}}$$

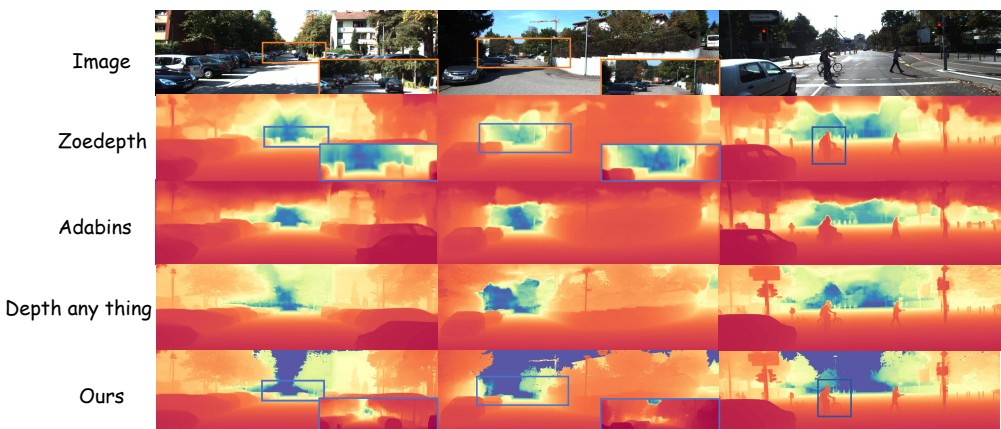

Figure 8: **Qualitative comparisons of depth predictions on the outdoor dataset KITTI**. It can be observed that our method performs better in predicting details both in the near range and far distance. To improve visualization clarity, we cropped the part beyond 80 meters, but we still achieve good predictions for that portion.

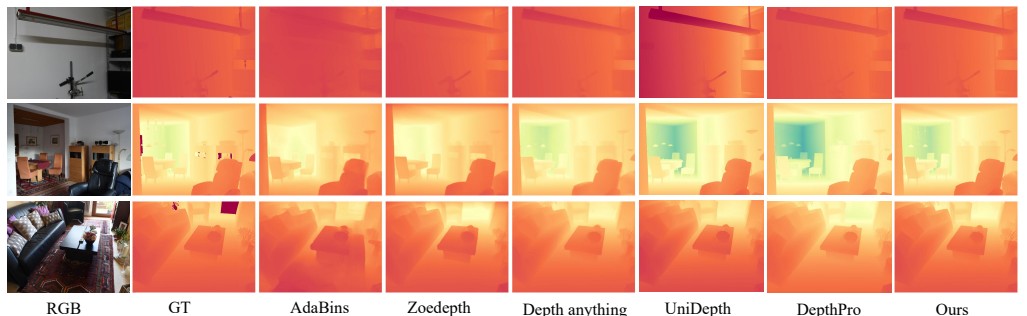

Figure 9: **Qualitative comparisons of depth predictions on the indoor dataset iBims**. It can be observed that our method performs better in predicting details in the near range (e.g., the first row). Moreover, our predictions for long distances in indoor scenes are more accurate (e.g., the second row).

Average $\log_{10}$ Error:

$$\log_{10} = \frac{1}{M} \sum_{i=1}^{M} \left| \log_{10} \mathbf{d}_i - \log_{10} \hat{\mathbf{d}}_i \right|$$

Threshold Accuracy ($\delta_n$): The percentage of pixels such that

$$\max \left( \frac{\mathbf{d}_i}{\hat{\mathbf{d}}_i}, \frac{\hat{\mathbf{d}}_i}{\mathbf{d}_i} \right) < 1.25^n \quad \text{for } n = 1, 2, 3$$

Here, $\mathbf{d}_i$ and $\hat{\mathbf{d}}_i$ denote the ground-truth and predicted depths at pixel $i$, respectively, and $M$ is the total number of valid pixels in the image.

We cap the evaluation depth at 10 meters for indoor datasets and 80 meters for outdoor datasets. Final predictions are obtained by averaging the depth map of the original image with that of its horizontally flipped counterpart, and evaluations are performed at the original ground-truth resolution.

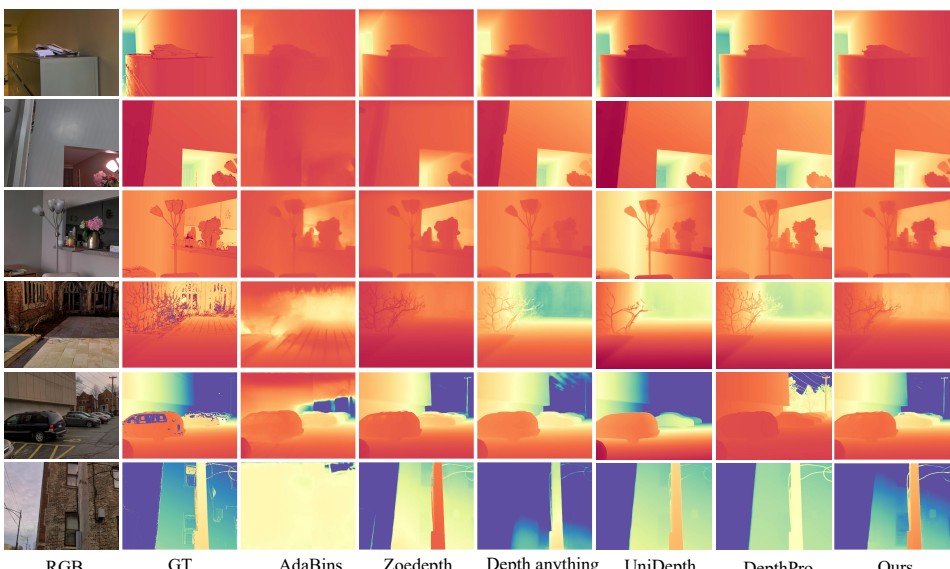

Figure 10: **Qualitative comparisons on diode**. Our test results on the DIODE dataset, which is a real-world dataset containing both indoor and outdoor scenes, demonstrate that our method achieves high accuracy in mixed scenes.

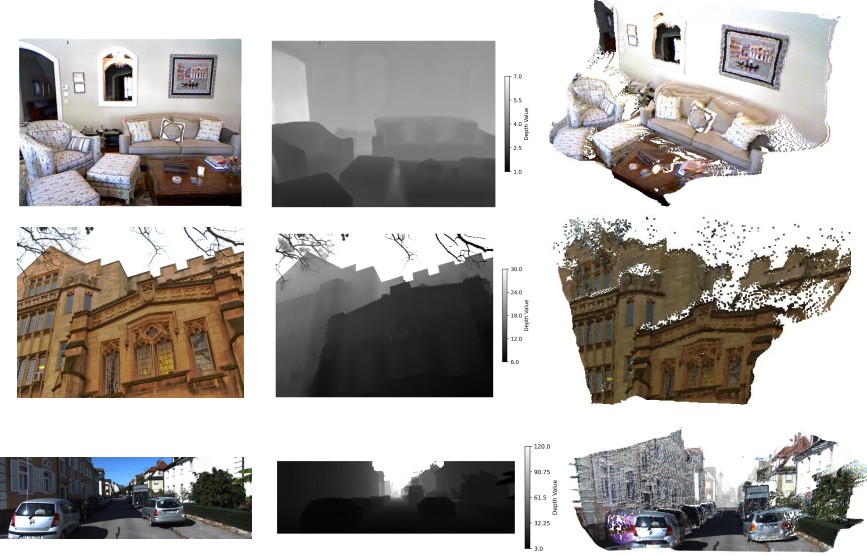

Figure 11: **3D Reconstruction based on Depth Estimation** We obtain the metric depth of different scenes through depth estimation and acquire the point cloud information of the scene via back-projection. This allows us not only to obtain an accurate scale of the scene but also to achieve absolute depth estimation across different scenes.

## H  APPLICATION - MONOCULAR RECONSTRUCTION

In monocular surface reconstruction, we use our predicted metric depth maps to directly reproject the 2D depth information into 3D space. By utilizing the camera intrinsics, we can convert the depth values to real-world coordinates. This allows us to construct a 3D point cloud of the scene from a single image. Our approach's ability to provide accurate and unified depth predictions across varying

scene scales makes it useful for reconstructing detailed 3D structures from monocular images, even in scenes with significant depth variations, such as indoor and outdoor environments. Please refer to Fig. 11 in the supplemental for more details.

