# OpenReview forum: "Metric-Solver: Sliding Anchored Metric Depth Estimation from a Single Image"
_ICLR.cc/2026/Conference — ICLR 2026 Conference Withdrawn Submission_

### Official Review · Reviewer_VdRe · 2025-10-28

**Soundness:** 2
**Presentation:** 3
**Contribution:** 2
**Rating:** 4
**Confidence:** 5

**Summary:**

This article proposes Metric-Solver, a novel sliding anchor-based monocular metric depth estimation method, to address challenges of diverse depth scales in indoor/outdoor scenes. It uses a reference anchor to split scene depth into scaled near-field (linear normalization) and tapered far-field (exponential normalization) components, enabling unified representation of depths from 0 to infinity. The anchor slides along the depth axis to adapt to inter/intra-scene variations, enhancing precision for near objects and stability for far regions.

**Strengths:**

**Efficient architecture**: Adopts "one shared DINOv2 encoder + two lightweight decoders"—shared features reduce redundancy, and lightweight decoders ensure efficiency. Only adds ~58M parameters and ~4ms inference time vs. DepthAnything, supporting 125 FPS real-time inference.

**Strong generalization**: Outperforms baselines in zero-shot tests across 4 indoor/outdoor datasets (e.g., iBims, DIODE) and achieves SOTA on NYU-V2/KITTI. No need for scene-specific assumptions (e.g., fixed max depth), showing robust cross-dataset adaptability/

**Weaknesses:**

1. **Incomplete Computational Efficiency Comparison**
The paper presents efficiency metrics (parameters, inference time, memory) in Table 4 but lacks two key elements: a direct comparison with the baseline ZoeDepth (a frequently referenced method in metric depth estimation) and FLOPS data—critical for evaluating computational overhead. Adding these would help readers assess if performance gains are balanced with practical deployment costs.
2. **Underutilization of Sliding Anchors in Inference**
To ensure fair baseline comparisons, the sliding anchor mechanism is fixed (e.g., 0.5 threshold for masks) during inference. While reasonable for consistency, this limits the mechanism’s core advantage—dynamic adaptation to scene scales. Exploring scene-adaptive anchor selection (e.g., via semantic cues) would better showcase its potential.
3. **Insufficient Depth Scaling and Supervision Details**
The paper uses linear normalization for near-field depth and exponential normalization for far-field depth, but the rationale for these choices (e.g., why not other functions) is underdiscussed. Additionally, experimental error distributions across near/far ranges and theoretical derivations for depth continuity need more refinement to strengthen rigor.
4. **Unexplained Performance Disparity on iBims**
Metric-Solver performs notably better on the iBims dataset than others (e.g., DIODE, SYNTHIA), but the paper provides no context. This gap weakens the persuasiveness of its generalization ability.
5. **Ambiguous Core Contribution of Sliding Anchors**
Prior work (ZoeDepth, AdaBins, DualDepth) has explored near/far-field separation and dual-branch fusion. Ablation experiments suggest the two-decoder architecture drives more performance gains than the sliding anchor—making the latter’s unique value unclear. .
6. **Undetailed Anchor Embedding Design**
How are anchor  embeddings initialized? Only channel-wise concatenation is used for feature fusion, with no testing of alternatives (e.g., adding positional encodings)—limiting the comprehensiveness of this design.
7. **Unaddressed Depth Continuity for Large-Span Objects**
For objects with large depth spans (e.g., approaching trains, full walls), rigid anchor-based scaling may cause unnatural depth transitions. The paper does not test this issue or propose fixes, which could affect practical application in complex scenes.

**Questions:**

Please see weakness

---

### Official Review · Reviewer_TyWr · 2025-10-31

**Soundness:** 3
**Presentation:** 3
**Contribution:** 2
**Rating:** 4
**Confidence:** 3

**Summary:**

This paper proposes a new approach for estimating a metric depth from a single image. Specifically, the authors introduce a sliding anchor-based representation that can adapt to diverse depth distributions in both indoor and outdoor scenes. The method allows flexible anchor selection during inference. Extensive experimental results support the validity of their work.

**Strengths:**

	Important task – metric depth estimation

	The proposed sliding anchor mechanism introduces an interpretable way to adjust depth range. Ablation studies suggest it contributes to performance.

	The approach can potentially benefit downstream applications requiring controllable metric depth estimation.

**Weaknesses:**

Major weaknesses are as below:

	The novelty of this paper seems marginal. The paper mostly extends existing depth estimation frameworks with sliding anchor-based control. To my understanding, the suggested idea is conceptually similar to Bi3D's range-selective scheme [1], with the primary difference being the monocular metric depth.

[1] Badki, Abhishek et al. "Bi3d: Stereo depth estimation via binary classifications." CVPR 2020.

	The paper does not clearly explain why their method outperforms existing approaches for zero-shot generalization experiments (Table 1). I am curious about how much of the reported performance gain actually comes from the sliding anchor mechanism itself, especially on the indoor dataset.

	While the authors emphasize that users can interactively select or slide the anchor during inference with some suggested anchor selection strategies, there is no clear criterion for determining whether a chosen anchor is appropriate or incorrect for a given scene. Moreover, the experiments appear to fix 10m for indoor and 80m for outdoor, which is domain-specific.

Overall, I am not fully convinced that the proposed method is powerful compared to existing works, and suspect that this anchor-based depth representation is truly dynamic as claimed. I will reconsider the score when these concerns are addressed well.

Some minor typo issues are as below:

	Line 177: randonly -> randomly

	Line 195: each components -> each component

**Questions:**

With the weaknesses mentioned above, I have a few more questions about the paper.

	For zero-shot generalization experiments, is the policy for selecting anchor value identical to that used for the NYU/KITTI comparisons, or domain-specific?

	In line 211, the authors choose a threshold of 0.5 to obtain the binary mask. Why does it work for both indoor and outdoor scenes?

	For practical usage, if the user can interactively select or slide the anchor, how can one determine whether the chosen anchor is appropriate or incorrect for a given scene?

---

### Official Review · Reviewer_fTp8 · 2025-11-01

**Soundness:** 2
**Presentation:** 2
**Contribution:** 2
**Rating:** 2
**Confidence:** 4

**Summary:**

The paper introduces a sliding anchor mechanism for monocular metric depth estimation. The method combines near- and far-depth decoding branches anchored by a variable depth reference to adaptively normalize depth across varying scales. By fusing two depth maps with a learned mask and leveraging a DINOv2 backbone, the approach aims to achieve better generalization across indoor and outdoor scenes without needing scene-specific calibration. Experimental results on standard benchmarks such as NYU-V2 and KITTI as well as zero-shot evaluation sets show competitive or superior performance relative to the prior works which are considered in the comparisons.

**Strengths:**

1. Novel architectural module for robust prediction of depth values with diverse ranges. The introduction of a sliding anchor mechanism is a creative attempt to address scale normalization in metric depth estimation, a long-standing challenge. The method’s two-branch decoding and anchor-based fusion represent a novel architectural variation within transformer-based depth networks.

2. Breadth of evaluation. The authors test across multiple standard in-domain as well as zero-shot datasets, demonstrating cross-domain applicability of the method.

**Weaknesses:**

1. Manipulated and incomplete presentation of the previous state of the art in the main experimental comparisons. In the main comparisons on zero-shot generalization of Table 1 and in-domain estimation of Table 2, the authors:
* have reported false figures on DIODE Indoor in Table 1 for both Depth Pro (Bochkovskii et al. 2025) and UniDepth (Piccinelli et al. 2024), invariably reporting a substantially reduced performance for both models compared to the actual figures provided by the respective papers. In particular, on DIODE Indoor, UniDepth originally reports $\delta_1 = 77.4$%, REL $= 17.2$%, and RMSE $= 0.954 m$, whereas the authors report for it $\delta_1 = 27.8$%, REL $=47.9$%, and RMSE $= 1.741 m$. Moreover, Depth Pro on DIODE Indoor originally achieves $\delta_1 = 67.1$%, REL $= 19.9$%, and RMSE $= 0.900 m$, whereas the authors report for it $\delta_1 = 40.1$%, REL $=36.3$%, and RMSE $= 1.462 m$. That is, both UniDepth and Depth Pro factually outperform the proposed method by a large margin on DIODE Indoor, but the false figures presented by the authors create the opposite picture.
* have omitted two central very recent state-of-the-art works from the zero-shot comparison of Table 1, i.e. UniK3D [A] and Metric3Dv2 (Hu et al. 2024a). This omission is important, as both of these works substantially outperform the presented method on sets which are included in the comparison of Table 2. More specifically, UniK3D achieves on IBims $\delta_1 = 91.9$%, REL $= 10.4$%, and RMSE $= 0.406 m$, outperforming the presented method in all examined metrics. In addition, on DIODE Indoor, Metric3Dv2 obtains $\delta_1 = 94.0$%, REL $= 9.3$%, and RMSE $= 0.399 m$ and UniK3D obtains $\delta_1 = 71.3$%, REL $= 16.1$%, and RMSE $= 0.767 m$, i.e. both outperform the proposed approach substantially.
* have omitted at least three central recent state-of-the-art works from the in-domain comparison of Table 2. i.e. UniDepth (Piccinelli et al. 2024), Metric3Dv2 (Hu et al. 2024a), and UniK3D [A]. For example, UniDepth outperforms the authors' method on KITTI, obtaining $\delta_1 = 98.6$%, REL $= 4.2$%, and RMSE $= 1.75 m$.

A highly inaccurate picture regarding the experimental performance of the method compared to the current state of the art is thus created due to the above figures manipulations and method omissions. The third main contribution in the respective list claimed by the authors in the end of the Introduction section is consequently unsupported.

2. No metric output. "Metric" originally means that the network's output lives in a metric 3D Euclidean space. However, the presented method only predicts a mere (metric) depth map, which lacks information for capturing the metric structure of the scene. Such capturing would require outputting the ray directions corresponding to the various pixels. Especially at zero-shot scenarios where intrinsics might not be given, this shortage prevents the proposed method from true metric prediction. This deficit is also evident by the absence of metric 3D evaluation measures, such as Chamfer distance or $F$-score, across the experimental evaluations of the paper. The authors could have at least used the ground-truth camera intrinsics provided with several of the evaluation sets they examine, in order to upgrade their metric depth maps to metric 3D maps and evaluate more meaningfully on 3D metrics against competing methods.

[A] Piccinelli et al.: UniK3D: Universal camera monocular 3D estimation. In CVPR, 2025.

**Questions:**

1. Can the authors evaluate their method for 3D-level metrics against all central competing state-of-the-art metric depth/3D estimation approaches? (Cf. Weakness 2)

**Details Of Ethics Concerns:**

Systematic manipulation of performance figures associated with competing state-of-the-art approaches in the main experimental comparison of the paper (Table 1), in a way that the reported figures in the paper are invariably substantially inferior to the figures originally provided by the respective previous works (cf. Weakness 1 in my review).

---

### Official Review · Reviewer_75q8 · 2025-11-02

**Soundness:** 2
**Presentation:** 2
**Contribution:** 2
**Rating:** 4
**Confidence:** 3

**Summary:**

The paper proposes a sliding-anchor representation for monocular metric depth. A single scalar anchor partitions depth into two ranges: a scaled-near head (linear normalization) and a tapered-far head (exponential normalization), fused via a learned mask. Conditioning on an anchor pool lets inference focus on near or far distances. The backbone uses a large-scale visual encoder with dual DPT-style decoders. training mixes indoor/outdoor real + synthetic data. Experiments report zero-shot and fine-tuned results against recent baselines.

**Strengths:**

1. The core idea of using anchors for scene generalization makes sense. Also, the split scaled-near and tapered-far, plus a learned fusion mask, gives a simple, reusable recipe for spanning both very close and extremely far depths without saturating either branch.

2. The method has good empirical results compared to current sota baselines.

**Weaknesses:**

1. The novelty is incremental. The sliding-anchor near/far split with mask fusion largely repackages known ideas (adaptive binning, log/inverse-depth compression, multi-head/gating). Without automatic anchor selection or a learned taper, it reads as an engineered variant rather than a fundamentally new formulation.

2. There is no automatic anchor selection. This makes the method more like a tunable tool than a fully automatic monocular metric depth estimator. The paper does not close the loop: it never shows a mechanism that chooses the correct anchor for a novel test image, nor does it quantify robustness to wrong anchor choice.

3.  The exponential taper constant k is fixed. This constant strongly influences far-range shape and gradients. Also, there is no discussion of stability.

4. Overclaiming SOTA despite mixed results: on the DIODE-outdoor, the proposed method loses to DepthPro, so "SOTA everywhere" is not correct.

**Questions:**

see above weakness

---

### Note · Authors · 2026-01-19

I have read and agree with the venue's withdrawal policy on behalf of myself and my co-authors.